# Profiling the Murine Acute Phase and Inflammatory Responses to African Snake Venom: An Approach to Inform Acute Snakebite Pathology

**DOI:** 10.3390/toxins14040229

**Published:** 2022-03-22

**Authors:** Jaffer Alsolaiss, Chloe A. Evans, George O. Oluoch, Nicholas R. Casewell, Robert A. Harrison

**Affiliations:** 1Centre for Snakebite Research and Interventions, Liverpool School of Tropical Medicine, Pembroke Place, Liverpool L3 5QA, UK; chloeevans1006@gmail.com (C.A.E.); nicholas.casewell@lstmed.ac.uk (N.R.C.); robert.harrison@lstmed.ac.uk (R.A.H.); 2Kenya Snakebite Research & Intervention Centre, Institute of Primate Research, Karen, P.O. Box 24481, Nairobi 00502, Kenya; ogemosh@gmail.com

**Keywords:** snakebite, envenoming, African cobra venoms, ex vivo human blood, in vivo murine model, acute phase proteins, inflammatory responses

## Abstract

Snake envenoming causes rapid systemic and local effects that often result in fatal or long-term disability outcomes. It seems likely that acute phase and inflammatory responses contribute to these haemorrhagic, coagulopathic, neurotoxic, nephrotoxic and local tissue destructive pathologies. However, the contributory role of acute phase/inflammatory responses to envenoming is under-researched and poorly understood—particularly for envenoming by sub-Saharan African venomous snakes. To provide data to help guide future studies of human patients, and to explore the rationale for adjunct anti-inflammatory medication, here we used an in vivo murine model to systematically assess acute phase and inflammatory responses of mice to ten African snake venoms. In addition to investigating snake species-specific effects of venom on the cardiovascular system and other key organs and tissues, we examined the response to intravascular envenoming by acute phase reactants, including serum amyloid A, P-selectin and haptoglobin, and several cytokines. Venoms of the spitting (*Naja nigricollis*) and forest (*N. melanoleuca*) cobras resulted in higher acute phase and inflammatory responses than venoms from the other cobras, mambas and vipers tested. *Naja nigricollis* venom also stimulated a 100-fold increase in systemic interleukin 6. Thin blood films from venom-treated mice revealed species-specific changes in red blood cell morphology, indicative of membrane abnormalities and functional damage, lymphopenia and neutrophil leukocytosis. Our ex vivo assays with healthy human blood treated with these venoms identified that *N. nigricollis* venom induced marked levels of haemolysis and platelet aggregation. We conclude that African snake venoms stimulate very diverse responses in this mouse model of acute systemic envenoming, and that venoms of the African cobras *N. nigricollis* and *N. melanoleuca*, in particular, cause marked inflammatory and non-specific acute phase responses. We also report that several African snake venoms cause haemolytic changes. These findings emphasise the importance of understanding acute responses to envenoming, and that further research in this area may facilitate new diagnostic and treatment approaches, which in turn may lead to better clinical outcomes for snakebite patients.

## 1. Introduction

Snakebite is a neglected tropical disease (NTD) annually inflicted upon over 2.7 million people, causing mortality in around 138,000 victims and leaving over 400,000 survivors with life-altering physical and perhaps psychological disabilities [1,2,3]. The burden of snakebite is greatest in the rural tropical regions of South and Southeast Asia, Latin America and sub-Saharan Africa [4]. In Africa, between 20–32,000 deaths are thought to occur each year, and this continent suffers the highest snakebite case fatality rate [2]. Venomous snakes within the Elapidae (e.g., cobras, mambas) and Viperidae (e.g., adders and vipers) families are considered the most medically important [2].

In Africa, envenoming by the *Echis* (saw-scaled or carpet vipers) and *Bitis* (puff adders) viper species causes local oedema that can lead to necrosis, systemic bleeding, coagulopathy and cardiovascular shock [2]. Envenoming by *Dendroaspis* mambas and non-spitting *Naja* cobras is typically associated with rapid, descending neuromuscular paralysis (slurred speech, ptosis that can proceed to respiratory paralysis) [2]. Neurotoxicity is rarely a consequence of envenoming by the African *Naja* spitting cobras and *Hemachatus* rinkhals snakes, which primarily cause local, rapidly progressive and painful swelling that can lead to necrosis. The type and severity of pathology caused by snake envenoming are dictated by the toxin composition of the venom and the amount of venom injected, both of which vary between species and genera [2,5], the location of the bite site and the size and health of the victim. Our understanding of the pathological course of envenoming derives from clinical observations of hospitalised patients, and from experimental animal studies focused primarily on the direct effects of venom toxins on tissues. We know remarkably little about the role of inflammatory and acute phase responses that occur during local and systemic envenoming.

Like other acute tissue injuries, an early effect of almost all snake envenomings is local and instant severe pain, heat, redness and/or swelling, often within two hours [6]. Rapid inflammatory responses to tissue injury are a fundamental part of the body’s defence system, intended to protect against injury, stimulate tissue repair and hinder the systemic spread of foreign bodies. It is the rapid release of numerous inflammation-mediators including alarmins, histamines, chemokines and cytokines (e.g., interleukin (IL) -1, IL-6, and tumour necrosis factor-alpha (TNF-α)) [7] that drive increased vascular permeability, vasodilation and chemotaxis, which ultimately facilitates the diffusion of plasma and leucocytes to the interstitial tissue [8]. Another non-specific systemic reaction to injury is the acute phase response (APR) consisting of rapidly elevated amounts of positive acute-phase proteins (e.g., C-reactive protein (CRP), serum amyloid A (SAA), fibrinogen and α-globulins), and reduced release of negative acute-phase proteins (e.g., albumin) [9]. Quantifying individual acute-phase proteins can therefore provide insight into the magnitude of the triggering event [9]. Other physiological events accompanying the APR to injury include increased levels of circulating leucocytes, activation of complement and blood coagulation cascades, and biosynthesis of adreno-corticotropic hormones [10,11].

Inflammatory responses in snakebite patients have been greatly under-researched. Stone et al. (2013) reported that patients envenomed by *Daboia russelii* (Russell’s viper; *n* = 113) exhibited elevation of anaphylatoxins and pro- and anti-inflammatory cytokines (IL-6 and IL-10) [10]. Avila-Agüero et al. (2001) observed that nine of eighteen victims of *Bothrops asper* envenoming exhibited elevated levels of IL-6, TNF-α and IL-8 [12]. Barraviera et al. (1995) detected elevated levels of IL-6 and IL-8 in victims of *Bothrops* (*n* = 16) and *Crotalus* (*n* = 15) envenoming, but no significant change in IL-1β in either type of envenoming [13]. Marked leucocytosis with neutrophilia and lymphopenia and a decline in albumin levels were also detected and, in three of four studied patients, a transient increase in CRP (the only CPP studied) was also noted.

Veterinary clinical studies of snakebite, including by African snakes [14,15,16,17,18], have provided some more detail on acute reactions to envenoming. Dogs envenomed by *Vipera berus* (European Viper), *Bitis arietans* (puff adder) and *Naja annulifera* (snouted cobra) all showed elevated levels of CRP at, and for 12 h after, admission [15,17]. In vivo experimental mouse studies also report acute phase responses to snake venoms. Examination of the wound exudate at the site of mice injected with *Bothrops atrox* (common lancehead) venom revealed an abundance of cytokines, chemokines and molecules associated with damage associated molecular patterns (DAMPs; derived from the proteolysis of proteins by venom toxins and host proteinases), whose activity is driven through interaction with toll-like receptors (TLRs). Mice pre-treated with a TLR4 antagonist exhibited reduced vascular permeability after venom injection [19]. Furthermore, wild-type, and not TLR2 gene knockout mice, injected with *B. atrox* venom had higher levels of IL-6 and monocyte chemoattractant protein-1 (CCL2), as well as an accumulation of polymorphonuclear cells [20]. These murine studies indicate that venom proteins can induce and modulate inflammation at the wound site through TLR4 and TLR2 signalling, respectively, and this signalling pathway offers some level of regulation. Other murine studies examining acute responses to systemic envenoming (summarised in Appendix A) are restricted to Latin American snake venoms—we could not find any information on acute phase and acute inflammatory responses to African snake venoms.

To address this knowledge gap, here we examined the acute responses in mice injected intravenously (representing systemic envenoming) with several medically important African snake venoms that cause distinct pathologies in human victims. However, since venoms inflict pain and distress to experimental animals, we also explored a 3R (Replacement, Reduction and Refinement) rationale by exploring the utility of in vitro tests that might mimic aspects of these in vivo toxicity assays. Various in vitro assays have previously been developed in the venom field with the aim of replacing and/or reducing the use of experimental animals, and examples include enzyme immunoassays for measuring treatment efficacy [21,22] and coagulation assays for profiling venom effects [23,24,25]. Here, we supplemented our in vivo systemic envenoming experiments with human ex vivo whole blood assays to assess whether our murine findings could be replicated in human blood spiked with these same African snake venoms.

## 2. Results

### 2.1. Haematological Analyses of Envenomed Mice

Blood samples to generate thin blood films were taken 6 h after iv venom injection with groups injected with a dose of venom approximating to one venom LD_50_ of each of the ten African snake venoms. Control thin blood films from PBS-injected mice (Figure 1A) provided normal cell morphology standards for comparison with thin blood films from the venom-injected mice.

#### 2.1.1. RBC Abnormalities Suggest Venom-Induced Haemolysis in Mice

The most pronounced pathology was the hyperchromic microspherocytosis (RBC whose diameter is less than normal, but thickness is increased) of mice subjected to *N. nigricollis* (Figure 1D) and *N. melanoleuca* venoms. This pathology was less severe in mice treated with six other venoms (both *Bitis* species, *E. ocellatus*, all three *Naja* species, and *D. viridis* and *D. jamesoni*) and absent from mice treated with *E. leucogaster* and *D. polylepis* venoms (Appendix A). These findings are indicative of red blood cell destruction, which can occur within the vascular system (intravascular haemolysis) or within the spleen or liver (extravascular haemolysis) and it may lead to anaemia or other dysfunctions. Ghost cells (post-haemolytic cell-like structure) were also observed in all cases exhibiting spherocytosis, suggesting a loss of haemoglobin into the extracellular space [26]. Mice subjected to *B. arietans* venom (Figure 1B) exhibited echinocyte (spiculated RBC) formation, which could perhaps be due to venom phospholipase-induced membrane abnormalities [27]. *D. jamesoni* (Figure 1C) venom caused dacrocyte (tear-drop RBC) formation, a manifestation associated with different forms of anaemia, including iron deficiency.

#### 2.1.2. Effects of Venom on Ex Vivo Human Blood

To relate these findings to venom activity on human blood, we incubated human red blood cells with the same venom doses used in the above murine experiments (Appendix A) to quantify the RBC haemolytic activity of each venom and the Hb content of venom-treated RBCs. All ten snake venoms caused a decrease in Hb compared to the control (RBCs incubated with PBS) (Figure 1E). While these reductions were relatively small (e.g., 86.9–97.7% Hb of the control), eight of ten venoms (*B. arietans*, *E. ocellatus*, *N. haje*, *N. melanoleuca, N. nigricollis*, *D. polylepis*, *D. viridis*, *D. jamesoni*) were found to cause statistically significant reductions (*p* ≤ 0.05; one-way ANOVA, Dunnett’s multiple comparisons test). For the haemolysis assay, PBS and Triton™ X-100 were used to generate negative and positive controls, respectively, representing 0% and 100% haemolysis (Figure 1F). *N. nigricollis* venom caused marked, significantly greater haemolytic activity (86.8%) than the negative control (*p* ≤ 0.05; one-way ANOVA, Dunnett’s multiple comparisons test). All other venoms exhibited a low level of direct haemolytic activity at these doses (1.7–5.3%), none of which were significantly elevated compared with the negative control (Figure 1F).

#### 2.1.3. *N. Nigricollis*, *N. Melanoleuca*, and *B. Arietans* Venoms Cause Neutrophilia in Mice

White blood cells (WBCs) are major components of the inflammatory process and total, and differential, WBC counts are often used as an early indicator of APR [28]. We analysed the mouse blood films at 40x objective for total WBC counts, and then at 100× objective to generate absolute, differential WBCs (Figure 2) for each group of mice. The control group had a mean total count of 12,400/μL with a distribution of different cell types within the reference range for mice (72–76% lymphocytes, 11–12% neutrophils, 7–9% monocytes, 6–7% basophils and <1% eosinophils). WBC counts were significantly reduced (*p* ≤ 0.05) in mice envenomed with *N. haje*, *D. polylepis, D. viridis, D. jamesoni* and *B. gabonica* venoms, with *N. haje* venom causing the greatest reduction (mean 3480/μL) (Figure 2A). With the exception of *D. viridis*, for which there was a high level of inter-group variation observed, all of the aforementioned venoms also caused a significant reduction in the number of lymphocytes (Figure 2B; *p* ≤ 0.05) compared to the control and, with the addition of *E. leucogaster* venom, all of these venoms also caused decreased basophil counts (Figure 2C; *p* ≤ 0.05). Contrastingly, the total WBC counts for *B. arietans*, *N. melanoleuca* and *N. nigricollis* envenomed animals were elevated (Figure 2A), as were their neutrophil counts, which were three to five times higher than control levels (Figure 2D; *p* ≤ 0.05). In the case of *N. nigricollis*, absolute monocyte counts were also elevated (Figure 2E; *p* ≤ 0.05). Eosinophil levels were notably elevated only in mice injected with *N. nigricollis* and *D. viridis* venoms compared to the control group (Figure 2F).

#### 2.1.4. Changes in Mouse Platelet Levels Following Injection with Viper and Elapid Venoms

Platelets are involved in inflammatory and haemostasis processes known to affect lymphocytes, neutrophils and monocytes [29]. Consequently, we examined the effect of venoms on platelet aggregation by analysing blood films from the envenomed mice. As murine platelets have a propensity to clump [30], we were unable to accurately estimate total platelet counts, as evidenced by our normal samples displaying atypically low counts. Instead, we qualitatively assessed the level of platelet clumping, with the results displayed in Table 1 (see also Appendix A for representative thin blood film images). The blood film of the normal control displayed platelet clumping with between two to three small colonies filed and each colony consisting of five to eight platelets. Blood films from the *N. melanoleuca, N. nigricollis*, and *D. jamesoni* envenomed mice showed three or more colonies per field and 11–20 platelets/colony. Contrastingly, *B. arietans and B. gabonica* venom appeared to potently inhibit platelet aggregation, as there was less than one small colony per field. Venoms from *E. leucogaster*, *N. haje, E. ocellatus, D. polylepis* and *D. viridis* caused smaller, perhaps non-pathological reductions in platelet aggregation.

#### 2.1.5. Platelet Aggregation in Ex Vivo Human Blood Exposed to Viper and Elapid Venoms

To assess the importance of these murine platelet aggregation findings to human envenoming, we treated ex vivo human platelets (separated from plasma) with the same amounts of venom and examined venom-induced platelet aggregation/inhibition (Figure 3). All venoms caused a change in platelet aggregation in comparison with the 20% platelet aggregation observed when platelet-rich plasma (PRP) was treated with collagen (positive control). For instance, *N. nigricollis* venom caused a significant increase (*p* ≤ 0.021 vs. collagen) in platelet aggregation followed by non-statistically significant increases caused by *N. haje, D. polylepis* and *E. ocellatus* venoms. The three remaining viper venoms (*B. gabonica*, *B. arietans* and *E. leucogaster*) caused significant inhibition (*p* ≤ 0.0001 vs. collagen) of platelet aggregation—a finding that supports our blood film analyses for these species.

### 2.2. Acute Phase and Acute Inflammatory Responses in Envenomed Mice

#### 2.2.1. P-Selectin Involvement in Acute Inflammatory Responses

Following an inflammatory stimulus, several molecules cause platelets and endothelial cells to become activated, leading to the surface expression and subsequent release into the systemic circulation of the transmembrane protein, P-selectin [31]. Consequently, and as a proxy for the level of endothelial cell and platelet activation, we quantified soluble P-selectin in the sera of envenomed mice by ELISA (Figure 4A). Mice injected with PBS had soluble P-selectin concentrations of 544 ± 84 ng/mL (mean ± SD), whereas those envenomed with *N. melanoleuca* and *N. nigricollis* venom exhibited around a two-fold increase (1197 ± 68 ng/mL and 907 ± 78 ng/mL, respectively; *p* ≤ 0.01 vs. control). At the other end of the spectrum, envenoming by *B. arietans* resulted in a significant decrease in P-selectin levels (147 ± 79 ng/mL; *p* ≤ 0.01 vs. control). The other venoms exhibited non-significant differences in P-selectin levels when compared with the PBS control.

#### 2.2.2. Envenoming-Related Immunoglobulin Responses

Pathogens and toxins are detected by circulating IgM antibodies. When an antigen-antibody complex forms, the classical complement pathway can become activated, heightening the antibody response [32,33]. Consequently, we next used a specific ELISA to quantify the levels of systemic IgM in sera of the envenomed mice (Figure 4B). We found that only treatment with *B. gabonica* venom led to marked increases (46%) in circulating IgM (*p* ≤ 0.05), while significant reductions in IgM were observed with *B. arietans* (66%)*, E. leucogaster* (60%)*, D. viridis* (60%) and *D. jamesoni* (66%) venoms (all *p* ≤ 0.05). No significant differences were found when comparing IgM levels caused by the other snake venoms in comparison with the control samples. These results need to be placed in the context of the 6-h duration of the experiment, and a longer period of exposure (not available to us in this study) with additional sampling time points would have been preferable.

#### 2.2.3. The Acute Phase Response in Mice Subjected to Viper and Elapid Venoms

The APR is an innate, non-specific, rapid systemic reaction to local or systemic disturbances. When stimulated, there are detectable and quantifiable changes in acute phase reactants from baseline levels: namely, positive acute phase proteins will increase and negative acute phase proteins will decrease [34]. In this study, we quantified levels of haptoglobin, SAA and CRP in mice envenomed with the 10 African snake venoms and compared these with non-envenomed controls. We observed five-fold increases in haptoglobin levels with mice dosed with *N. melanoleuca* and *N. nigricollis* venoms (*p* ≤ 0.0001 vs. control), while all other venoms caused haptoglobin levels to decrease (Figure 4C). While these reductions were modest, they were statistically significant in the case of *B. gabonica, B. arietans, E. leucogaster, D. polylepis* and *D. jamesoni*.

Only *N.*
*melanoleuca*
*venom* caused a significant (*p* ≤ 0.0001) increase in SAA levels (Figure 4D). We observed no significant changes in the levels of CRP across the different envenomed mice groups (data not shown for brevity), suggesting that CRP may not be a major acute phase protein in mice [35]. Although albumin appeared to decrease slightly in most groups treated with the various venoms, no significant reductions were identified when compared to the control (data not shown for brevity).

#### 2.2.4. Naja Nigricollis Venom Causes Rapid Release of Systemic Inflammatory Mediators

Cytokines are a diverse array of signalling peptides released by cells and are key regulators of the inflammatory and acute phase responses, including cell activation, proliferation, growth, differentiation, migration and apoptosis [36]. We investigated the levels of circulating cytokines in sera of mice injected with African viper and elapid venoms via multiplex analysis (Figure 4E). The PBS-only control mice exhibited undetectable levels of most of the cytokines investigated, although low levels of IFN-γ (2.08 pg/mL) and IL-1β (0.15 pg/mL) were observed. Mice subjected to N. *nigricollis* venom exhibited increased levels of several pro-inflammatory cytokines (IL-6, IL-18, IL-13, TNF-α, IL-5 and less marked increases of GM-CSF and IL-4). However, of these, significant elevations were only detected for IL-6 (*p* ≤ 0.0001 vs. control), where a dramatic increase was detected (119.57 ± 4.6 pg/mL vs. 0 pg/mL). Barely detectable elevations of IL-6 were noted from mice envenomed with *E. leucogaster*, *E. ocellatus*, *N. haje*, *N. melanoleuca* and *D. polylepis* venoms and their pathophysiological significance is therefore unknown. IFN-γ levels were elevated only in mice subjected to *D. polylepis* venom (other venoms provoked barely detectable increases). No statistically significant differences in TNF-α, IL-1β, IL-13, IL-5, GM-CSF, IL-12p70, IL-2, Il-4, IL-18 and IFN-γ levels were caused by any of the ten snake venoms. As outlined above, the short duration of murine in vivo experiments are likely limiting informative interpretations of cytokine responses to envenoming.

### 2.3. Serum Biochemistry in Venom-Injected Mice

Blood biochemistry analysis is a fundamental diagnostic indicator of injury to muscle and other organs. Consequently, we assessed a range of biochemical parameters to gain insight into the acute pathologies induced by this diverse group of ten African snake venoms in mice (Figure 5). Levels of bilirubin, a breakdown product of RBCs during haemolysis [37], were significantly raised in mice treated with all venoms except from *N. haje* and *N. melanoleuca* (Figure 5A). Only mice subjected to *B. arietans* venom exhibited a reduction in levels of total protein (Figure 5B) and albumin (Figure 5C), suggesting this venom is likely capable of inducing acute liver and kidney damage. Similarly, only *N. nigricollis* venom resulted in significant increases in levels of AST (Figure 5D) and ALT (Figure 5E), which are markers of hepatocellular damage. In combination with the significantly elevated levels of LDH (an indicator of haemolysis; Figure 5F) and CK (indication of muscle injury; Figure 5G), these findings suggest that *N. nigricollis* venom also caused erythrocytic, cardiac and renal damage in this mouse model of acute systemic envenoming. The raised levels of creatinine (Figure 5H) and LDH in mice treated with *N. melanoleuca* venom suggest this venom is also capable of causing renal and erythrocytic injury, in a comparable manner to *N. nigricollis* venom. Aside from bilirubin levels, the venoms of *B. gabonica*, *E. leucogaster*, *E. ocellatus*, *N. haje* and the three *Dendroaspis* spp. had little effect on the serum biochemistry of the envenomed mice under the experimental conditions used here (Figure 5).

## 3. Discussion

Snake venoms are extremely diverse in their composition, and they vary both interspecifically and intraspecifically [5,38]. It is this variation in venom composition that mediates the diverse array of clinical symptoms presented by snakebite victims [2]. Our murine and human results on the haematological changes caused by, and acute phase and inflammatory responses to, exposure with venom from ten different African snake species also reflect this complex interaction of venom with different mammalian physiological compartments. Despite well-known differences in venom composition among the different genera of snakes studied here, for example, the metalloproteinase-rich venoms of *Echis* vipers versus the three finger toxin or dendrotoxin rich venoms of cobras and mambas, respectively [5,39,40,41,42], we sought but could not find patterns linking each snake genera to distinct haematological, acute phase and acute inflammatory responses. Instead, as illustrated in Table 2, changes of these parameters in mice injected with venom or human blood products (red blood cells/platelets) exposed ex vivo to venom injection appeared very species-specific. Our discussion will therefore focus upon relating key findings from this study to human envenoming.

### 3.1. Inflammatory Cytokine Responses to Venom Injection

In snakebites, the extent of local necrosis is significantly affected by the release of cytokines such as IL-1 and TNF-α induced by snake venom metalloproteinases (SVMP). Thereafter, these cytokines activate the endogenous metalloproteinases in various cells (e.g., fibroblasts), which can cleave TNF-α and amplify the process of cell destruction and necrosis [43]. Despite the clinical relevance of cobra (*Naja* spp.) envenoming in Africa, there have been few studies on the pathogenesis of inflammation-associated local tissue damage caused by such species [44,45] and there is a paucity of knowledge on the systemic response to these venoms. IL-6, TNF-α and IL-1β are known to be the primary mediators of the APR [46,47]. While one study on serum IL-6 levels in human snakebite victims identified elevated IL-6 levels correlated with non-specific clinical symptoms (nausea, vomiting, headache, abdominal pain, diarrhoea) in one case [10], other studies were unable to identify a significant clinical association between elevated serum cytokine concentration and the severity of envenoming [12]. Additionally, in a murine model, anti-cytokine antibodies (anti-TNF-α, anti-IL-1β anti-IL-6) failed to reduce lethality, haemorrhage, necrosis or oedema in mice exposed to *Bothrops* venom. Our murine results with African venoms (Figure 4) largely concur with this lack of involvement of key inflammatory cytokines with acute envenoming—with the striking exception of *N. nigricollis* venom causing dramatically elevated levels of serum IL-6, while serum levels of TNF-α and IL-1β remained close to the baseline. The black mamma venom (*D. polylepis*) was the only venom to cause significant elevation of IFN-γ. These findings may well reflect the short time duration of experimental envenoming applied in this study (six hours), and future research over longer time courses would likely be informative.

### 3.2. Leukocyte Changes to Venom Injection

Once cytokines become systemically distributed, they can act on the bone marrow to modify leucocyte production, and IL-6 has been shown to be important in leucocyte recruitment and neutrophil migration during the inflammatory response [48]. We found that within only 6 h after injection, both *N. nigricollis* and *N. melanoleuca* venoms caused significant changes to the circulating leukocyte profile, with a significantly higher ratio of neutrophils to lymphocytes compared to the control. This response is typical of acute stress syndrome and inflammation, and while this has previously been reported in human snakebite victims following viper envenoming [13,49], it is unreported for elapid envenoming. The number of leukocytes and their subtypes has been recognised as a marker of inflammation in disease [50] and the neutrophil:lymphocyte ratio has been reported as a risk factor for the severity of the outcome of snakebite [51,52]. Aktar and Tekin (2017) investigated several parameters in 142 Iranian children with snakebite, and those that were grouped as ‘severe’ had significantly higher leucocytes and neutrophils than the other groups [53]. The clinical symptoms of these children were distinct in that they suffered extensive swelling extending to the trunk, necrosis or compartment syndrome, as well as clinical signs of systemic effects (systemic bleeding, hypotension, disseminated intravascular coagulation (DIC) or renal failure, cerebral haemorrhage, or multi-system failure).

### 3.3. Haptoglobin and Serum Amyloid A Responses in Mice Injected with Venom

Another fundamental part of the acute phase response occurs when inflammatory mediators stimulate hepatocytes to alter the production of certain proteins. Haptoglobin is an acute phase protein in humans and mice, and although its key role is to bind free haemoglobin following haemolysis to prevent oxidative damage [54], haptoglobin will also increase rapidly in response to infection and/or injury, exhibiting a range of immunomodulatory properties [55,56]. Here, in this mouse model of acute systemic envenoming, we identified physiologically relevant elevations in haptoglobin in mice injected with *N. melanoleuca* or *N. nigricollis* venoms. Contrastingly, haptoglobin levels in mice injected with *B. gabonica* and *B. arietans* viper venoms were significantly lower than the normal control, suggesting a predominantly haemolytic response. Low levels of haptoglobin have previously been used as a marker of coagulopathy in snakebite victims [57], a pathology not often associated with *B. gabonica* and *B. arietans* envenoming [58]. Our observation that mice injection with *B. gabonica* venom responded with elevated IgM and decreased serum albumin levels may be linked to the report that these concomitant changes are a characteristic of endothelium disturbances induced by haemotoxic snake venoms, allowing fluid to leak into the interstitial space and contributing to oedema [59]. Although SAA is also a major acute phase protein, it is produced in a time-dependent manner and typically reaches its peak after 24–72 h. We found SAA to be elevated only in mice injected with *N. melanoleuca* venom in the shorter time frame of our study.

### 3.4. Venom-Induced Changes in Markers of Renal, Hepatic, Muscle and Blood Cell Damage

Biochemical analyses of serum also indicated that mice envenomed with *N. nigricollis* and *N. melanoleuca* venoms showed unique pathological characteristics in comparison with the other venoms tested. Both groups had significantly elevated LDH, which is indicative of intravascular haemolysis [60], and marked spherocytosis was apparent in the blood film analysis. Spherocytes are damaged cells that are highly susceptible to destruction. Typically, they can be found in all haemolytic anaemias to some degree—a medically relevant consequence of snakebite [61]—and occur following complement and/or immunoglobulin fixation (immune-mediated haemolytic anaemia (IMHA)) or due to direct damage [62]. Secondary IMHA has been reported in dogs following envenoming by the tiger snake, *Notechis scutatus,* based on similar diagnostic parameters to those included in this study (e.g., evidence of haemolysis (ghost cells), marked spherocytosis, and raised AST, ALT and total bilirubin) [63]. Our findings support the possibility that IMHA is the direct result of the action of venom toxins. Furthermore, Tambourgi et al. (1994), who performed in vitro haemolysis assays using sheep RBCs, reported that both *N. nigricollis* and *N. melanoleuca* venoms caused haemolysis [64]. Our results of human red blood cells spiked ex vivo with venoms identified that *N. nigricollis*, but not *N. melanoleuca* venom, caused significant haemolysis at the venom doses tested.

Analysis of serum from mice injected with *N. nigricollis* venom also showed dramatically elevated (ten-fold above normal) CK levels—this elevation exceeds the five-fold threshold used as a clinical indicator in humans for the presence of rhabdomyolysis [65]. These findings were also supported by elevations, in the same experimental animals, of LDH—a marker of muscle breakdown [66]. Rhabdomyolysis is a major pathology that can occur following bites by a variety of elapid and viperid snakes and can lead to the accumulation of muscle breakdown products that cause kidney damage [67]. Vikrant et al. (2017) investigated acute injury in 121 snakebite victims in India and found intravascular haemolysis and rhabdomyolysis to be the most important risk factors for developing acute kidney injury [68]. Additionally, those who died (9.1%) had elevated WBC counts, elevated bilirubin and low levels of albumin compared to the surviving group—all of which were identified in the *N. nigricollis* envenomed mice used in this study. Furthermore, elevations in ALT, AST and total bilirubin in the *N. nigricollis* envenomed mice are also indicative of acute hepatocellular damage, a finding which is in line with the findings of James et al. (2013) who used rat models to investigate pathology caused by *N. nigricollis* venom. Liver injury is to be said one of the most common and serious symptoms of envenoming by Asian cobras (*Naja* spp.) [69].

Thrombosis and haemostasis are vital cellular processes, and different snake venoms disrupt these processes in a variety of ways. Thrombocytopenia, which is a reduction in the platelet count, is a common clinical observation in snakebite victims and is thought to be caused by venom-induced hyper-aggregation of platelets, particularly in those bitten by vipers [49,70,71,72]. However, a wide variety of viper venom toxins have been demonstrated to interact with platelets, and act via various mechanisms to either inhibit or promote their aggregation [73]. Against expectations, results from our model of acute systemic envenoming, and the ex vivo human plasma assay, identified that some elapid venoms caused the greatest levels of platelet aggregation, whilst viperid venoms appeared to minimise platelet aggregation (*B. arietans* and *E. leucogaster*). Platelets from humans are also the major source of soluble P-selectin [74], which is shed into the circulation to dampen the inflammatory potential of the cell. An in vitro study using human plasma also found these cobra venoms to have potent anticoagulant effects [75]. Here, the injection of mice with both *N. melanoleuca* and *N. nigricollis* venoms caused elevated levels of circulating P-selectin, suggesting high levels of platelet activation and endothelial dysfunction. A study by Sun et al. (2016) reported that P-selectin is physiologically involved in the neutralisation of venom-induced coagulopathy following *Crotalus atrox* envenoming in a mouse model [76].

We found that *B. arietans* venom caused significantly reduced P-selectin levels compared to the control and, as such, we propose that the venom somehow interferes with the function and/or activation of platelets, preventing the shedding of P-selectin. These findings also correlate with our blood film analysis, which revealed that *B. arietans* caused markedly decreased levels of platelet clumping compared with the controls. Indeed, it is well-established that *B. arietans* and *E. ocellatus* venom contains several anti-platelet peptides, including arietin (from *B. arietans*) and echistatin (from *E. ocellatus*), which suppress platelet aggregation by interacting with fibrinogen receptors on platelet membranes [77,78]. Furthermore, platelet aggregation has previously been shown to be strongly suppressed by whole *B. arietans* venom in vitro [79] and in humans [60,71].

## 4. Conclusions

We report that African snake venoms stimulate very diverse responses in this mouse model of acute systemic envenoming and that the venom of two African cobra venoms in particular, *N. nigricollis* and *N. melanoleuca*, cause marked inflammatory and non-specific acute phase responses. We also report venom-induced haemolytic activity in several cases of envenoming. We find that SAA and haptoglobin show promise as inexpensive, readily available clinical biomarkers of severe systemic envenoming. Despite these novel findings, our study has a number of limitations, most notably restrictive profiling of inflammatory and acute phase markers to a single time point following intravenous venom delivery in small numbers of experimental animals. Future research would benefit from time-course experiments over longer experimental durations (e.g., 24–72 h) and exploring alternate routes of venom delivery that may better mimic a real-world snakebite. Furthermore, there is a pressing need for similar research to be performed in human snakebite victims due to the potential for species-specific effects. Nevertheless, the findings presented herein emphasise the importance of understanding the acute responses that manifest following envenoming, and that further research in this area may facilitate new diagnostic and treatment approaches, which in turn may lead to better clinical outcomes for envenomed patients.

## 5. Materials and Methods

### 5.1. Snake Venoms

Lyophilised venoms from African snake species were donated to LSTM by Biological E. Limited, India as part of a separate venom-toxicity and antivenom-efficacy study. The venoms selected for use in this study were representatives of the four most medically important venomous snake groups found across Africa, namely the *Bitis* adders, *Echis* vipers, *Naja* cobras and *Dendroaspis* mambas. In total we used venom from four vipers: *Bitis gabonica* (Gaboon viper), *B. arietans* (puff adder), *Echis ocellatus* (West African saw-scaled viper) and *E. leucogaster* (white-bellied carpet viper), and six elapids: the cobras *Naja haje* (Egyptian cobra), *N. melanoleuca* (forest cobra) and *N. nigricollis* (black-necked spitting cobra), and the mambas *Dendroaspis polylepis* (black mamba), *D. viridis* (Western green mamba) and *D. jamesoni* (Jameson’s mamba). Lyophilised venoms were stored at 4 °C until reconstituted in phosphate-buffered saline (PBS, pH 7.2) for use, whereafter they were stored at −20 °C short-term (5–8 days).

### 5.2. Animals and Research Design

Murine in vivo experiments were undertaken under the UK Animals (Scientific Procedures) Act 1986 using protocols specified in Home Office grant Project Licence (#P24100D38) reviewed and approved by the Animal Welfare and Ethical Review Boards of the Liverpool School of Tropical Medicine and the University of Liverpool.

It is important to note that all results presented here were from samples obtained from mice used in venom-toxicity experiments—these experiments, therefore, contributed to both antivenom development and increasing knowledge on the systemic pathology of these venoms and the acute responses they induce. As part of the venom-toxicity study, and in line with World Health Organization recommended guidelines [80], the median lethal dose (LD_50_) for each venom was determined by intravenously (tail vein) injecting different venom doses in 100 μL volumes of PBS into groups of five male CD-1 mice (20–22 g; Charles River, UK). At the end of the experiment, the number of surviving mice injected with the dose of venom that corresponded closest to one venom LD_50_ (the amount of venom resulting in the survival of 50% of the injected mice; see Appendix A) were selected for sample collection to provide samples of comparable severity of systemic envenoming (*n* = 2 or 3; see Appendix A for details). A control group (*n* = 2) received 100 μL of PBS instead of venom. Throughout, experimental animals were continuously monitored for signs of severe envenoming and immediately euthanised at the first sign of severe cardiovascular or neurological distress (using a series of humane end points, such as paralysis or overt bleeding). Survival was recorded six hours post-envenomation and surviving animals euthanised. Throughout, euthanasia was performed using rising concentrations of carbon dioxide. Immediately after euthanasia, blood samples were collected via cardiac puncture. The blood from surviving mice from each group was used to prepare:a thin blood film (*n* = 50) for microscopic analysissera—blood was allowed to clot at room temperature, centrifuged for 10 min (9600× *g* at 4 °C) and sera stored at −20 °C (*n* = 50) to quantitate markers of:
oacute inflammatory responsesoacute phase responsesoliver, heart and kidney damage.

### 5.3. Blood Film Preparation and Haematological Analysis

Thin blood films prepared from murine blood were left to air-dry overnight, fixed for 60 s in methyl alcohol and stained with 10% Giemsa. A Leica ICC50W microscope was used to (i) morphologically profile the blood cells, (ii) count leucocytes, erythrocytes and platelets in the cell monolayer (10 fields) and (iii) record cell irregularities, such as erythrocyte agglutination or platelet aggregation. As the platelets were often aggregated, we were unable to generate accurate total platelet counts. To qualitatively assess the levels of platelet clumping we, therefore, used the following grading system: (i) no aggregation, <1 small colony/field; (ii) mild aggregation, 2–5 colonies/field and/or small colonies only; (iii) moderate aggregation, 2–8 colonies/field and/or small and medium colonies; (iv) marked aggregation, >5 colonies/field and/or medium and large colonies present. Small, medium and large colonies were defined as 2–10 platelets, 11–20 platelets and >20 platelets, respectively.

### 5.4. Quantification of Murine C-Reactive Protein and Cytokines by Multiplex Bead Array

The CRP Mouse ProcartaPlex™ Simplex Kit was used for serum CRP analysis, while the Th1/Th2 Cytokine 11-Plex Mouse ProcartaPlex™ Panel (both ThermoFisher Scientific, Vienna, Austria) was employed for serum cytokine analysis. All samples and standards were run in duplicate using manufacturer’s instructions with minor modifications, as follows: (i) Universal Assay Buffer (1×) was used for three-fold dilutions of the sera samples for the CRP simplex immunoassay, but the samples were not diluted for the cytokine multiplex immunoassay, (ii) every incubation step involved shaking the 96-well plate at 220 rpm, and (iii) after the beads, serum, standards and PBS were added, the mixture was left to incubate at 4 °C overnight. Luminex^®^ 100/200 technology (Mag-Plex^®^-Avidin Microspheres) was used for measuring bead fluorescence, and if the count did not exceed 100 or was not within the identifiable range, the data were excluded from downstream analysis. ProcartaPlex Analyst software was used for the analysis of results and the mean and standard deviation for each group was calculated.

### 5.5. Quantification of Serum Amyloid A, P-Selectin and Murine IgM by ELISA

The PHASE™ SAA Murine Assay Kit (Tridelta Development Limited, Ireland), P-selectin Mouse ELISA Kit (ThermoFisher Scientific, Vienna, Austria) and Mouse IgM ELISA Ready-SET-Go!™ Kit (Invitrogen™ eBioscience™, ThermoFisher Scientific, Vienna, Austria) were used in accordance with manufacturers guidelines to quantify serum amyloid A (SAA), P-selectin and murine IgM. Sera dilutions were performed in the supplied buffers for SAA, P-selectin and IgM at 1:200, 1:500 and 1:20, respectively. All samples were analysed in duplicate. Resulting absorbance measurements were taken at 450 nm (SAA, P-selectin) and 570 nm (murine IgM) using an LT-4500 microplate absorbance reader (LabTech International, Uckfield, UK). The mean values and standard deviation for each group were calculated.

### 5.6. Quantification of Murine Haptoglobin by Colorimetric Assay

We used the colorimetric PHASE™ Haptoglobin Assay (Tridelta Development Limited, Ireland) to quantify serum haptoglobin using the manufacturer’s instructions. The supplied stock calibrator and diluent were used for manual preparation of haptoglobin standards. Serum samples were not diluted before assessment and each sample was run in duplicate. After the samples were incubated at RT with chromogen for five minutes, the absorbance was read at 620 nm with an LT-4500 automatic microplate absorbance reader. For each group, the mean and standard deviation was calculated.

### 5.7. Standard Biochemical Analysis of Mouse Serum Samples

To quantify markers related to liver function, cardiac damage and renal function, we used an automated clinical chemistry analyser (Roche cobas c 701, Roche Diagnostics, Basel, Switzerland) using the manufacturer’s instructions. All enzymatic and colorimetric reagent kits, calibrators and controls were supplied by Roche Diagnostics UK Ltd. The mouse sera were diluted 1:20 before analysis in duplicate, alongside non-envenomed sera controls. Recording of the following analytes was performed using colorimetric and enzymatic methods: (i) liver function tests; alanine aminotransferase (ALT), total protein (TP), albumin and total bilirubin (TB); (ii) cardiac enzymes; creatine kinase (CK), lactate dehydrogenase (LDH), aspartate aminotransferase (AST); (iii) renal function tests; creatinine (CRT).

### 5.8. Collection of Human Blood for Ex Vivo Assays

Blood samples were obtained according to ethics-approved protocols (LSTM research tissue bank, REC ref. 11/H1002/9) from consenting healthy volunteers who confirmed they had not taken any anticoagulant treatments for at least three months prior to blood collection. Blood samples were collected in tubes containing different anticoagulants (acid citrate dextrose adenine [ACD-A] and ethylenediaminetetraacetic acid [K3-EDTA]).

#### 5.8.1. Quantifying Venom Haemolytic Activity on Human Erythrocytes

Blood was collected in K3-EDTA and 100 μL of red blood cells (RBCs) were first washed three times with 3 mL of PBS. Next, 100 μL of washed RBCs were resuspended with 3 mL PBS and added (100 μL/well) to 96-well microtiter plates (Nunc, ThermoFisher Scientific, Roskilde, Denmark). Next, the RBCs in each well were overlaid with 5 μL of venom samples equivalent to the LD_50_ dose of each venom used in the aforementioned murine in vivo experiments (Appendix A) in duplicate. These venom doses were selected to explore whether aspects of the in vivo findings could be replicated in human ex vivo experiments. The plates were incubated for 3 h at 37 °C and then centrifuged for five minutes at 1000× *g*. Subsequently, 80 μL of the supernatants were transferred to a new 96-well microtiter plate (Nunc) and an LT-4500 automatic microplate absorbance reader was used to generate haemolysis readings at 405 nm. The mean values and standard deviation were calculated.

#### 5.8.2. Quantifying Venom-Induced Aggregation of Human Platelets

To separate platelets, plasma and other blood components from human blood, we used a standard platelet-rich plasma (PRP) separation technique [81], whereby Acid Citrate Dextrose-Solution A (ACD-A) blood samples were subjected to an 11 min centrifugation step at 150× *g*. Following isolation, platelets were manually counted to confirm normal levels (350 × 10^9^ platelets per microliter of blood) [82]. For the platelet aggregation assay, 96-well microtiter plates (Nunc) were loaded with 72 μL/well platelet-rich plasma and incubated with 8 μL of each venom solution, which contained the previously defined murine LD_50_ dose as outlined above (and see also Appendix A), in duplicate. Next, we used a FLUOstar Omega microplate reader for kinetic measurement of the absorbance of the wells at 405 nm (40 min at 37 °C). Samples containing venom only and platelets only served as negative controls, while 72 μL of platelets incubated with 8 μL of 50 μg/mL human placenta collagen (Sigma-Aldrich^®^, Dorset Gillingham, UK) served as the positive control. Absorbance was plotted against time and platelet aggregation was measured by calculating the mean AUC of mean readings.

#### 5.8.3. Colorimetric Quantification of Venom-Induced Release of Human Haemoglobin

Cyanmethemoglobin techniques were employed for haemoglobin (Hb) measurements. First, a standard curve was prepared by diluting different concentrations of commercial Cyanmethemoglobin Standard Solution (180 mg/mL, 120 mg/mL, 60 mg/mL and blank) of appropriate haemoglobin in Drabkin’s Solution (used for quantitative, colorimetric determination of haemoglobin). The mixture was then incubated at room temperature for 5 min before absorbance was read at 540 nm using an LT-4500 automatic microplate reader (LabTech). The resulting absorbances were plotted to produce a standard curve (absorbance vs. cyanmethemoglobin concentration (mg/mL)). For venom experiments, human blood from healthy volunteers was collected in 4 mL ethylenediaminetetraacetic acid (K3-EDTA) tubes. Subsequently, the collected RBCs in EDTA were washed three times in PBS. Washed RBCs were then resuspended in PBS (1:1), and 100 μL aliquots were incubated with 100 μL of the LD_50_ dose (Appendix A) of each venom for 60 min at 37 °C. High (21 g/dL) and low (6.0 g/dL) controls of Hb (Sigma-Aldrich^®^, United Kingdom) and washed RBCs with no venom were also used. All samples were mixed before 10 μL of each sample was added to 625 μL Drabkin’s solution (Sigma-Aldrich^®^, United Kingdom) in duplicate and incubated for 20 min at room temperature. The resulting absorbances were measured at 540 nm using an LT-4500 automatic microplate reader. The formula from the standard calibration curve was used to quantitate the Hb levels in each sample.

### 5.9. Statistical Analyses

For all analyses, the significance of the differences of mean values between experimental groups was assessed by analysis of variance (ANOVA), followed by Dunnett’s multiple comparison test to compare means between experimental groups (samples treated with venom) and controls (no venom controls). A *p*-value of ≤0.05 was considered significant for all statistical tests. All statistical analyses were conducted using GraphPad Prism 8.0.

## Figures and Tables

**Figure 1 toxins-14-00229-f001:**
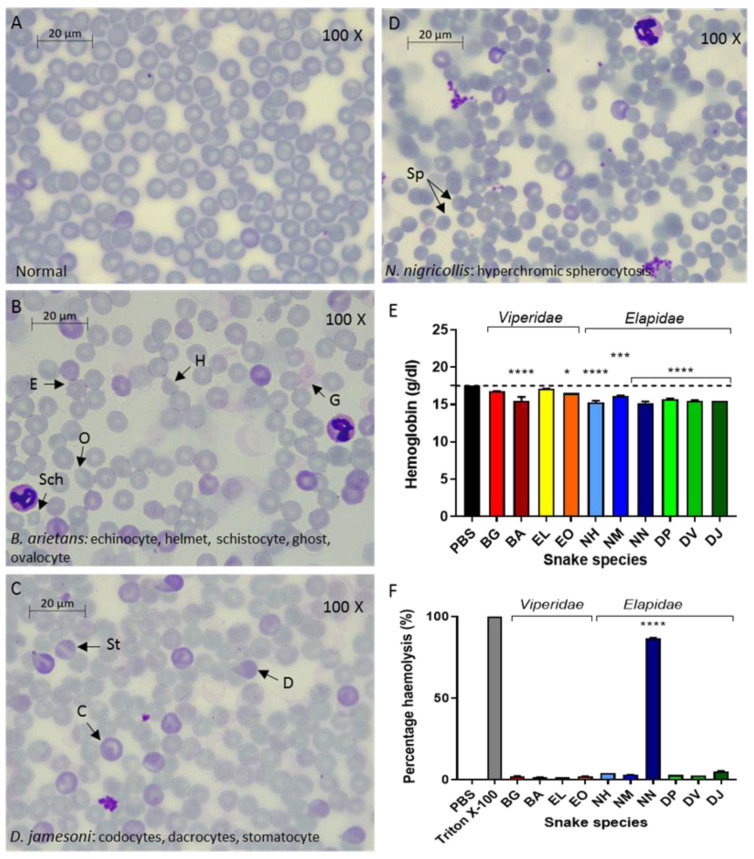
Normal (**A**) and abnormal murine red blood cell morphology (**B**–**D**) and haemoglobin content (**E**) and haemolysis (**F**) of human RBCs induced by snake venoms. Mice were injected with venom or PBS (normal) and after six hours samples were collected, and thin blood films were created. RBC morphology was observed at 100× objective. Representative blood films shown were from (**A**) non-envenomed control; (**B**) *B. arietans* venom, which caused RBC membrane abnormalities, illustrated by echinocytes (and helmet cells, and cell fragmentation (schistocytes (Sch)). Ghost cells were present, indicating loss of haemoglobin due to haemolysis; (**C**) *D. jamesoni* venom that caused codocyte, stomatocyte (St), and dacrocyte formation; (**D**) *N. nigricollis* venom caused hyperchromic microspherocytosis. Human RBCs were also incubated with the same LD_50_ dose of venom and used in ex vivo absorbance assays to determine: (**E**) haemoglobin levels and (**F**) the percentage of direct haemolysis, Key: Venoms; BG (*B. gabonica*), BA (*B. arietans*), EL (*E. leucogaster*), EO (*E. ocellatus*), NH (*N. haje*), NM (*N. melanoleuca*), NN (*N. nigricollis*), DP (*D. polylepis*), DV (*D. viridis*), DJ (*D. jamesoni*). Parameters from venom-incubation groups (*n* = 3) were compared to those from the negative control (PBS) with Dunnett’s multiple comparison test. Significant values are indicated by * *p* ≤ 0.05, *** *p* ≤ 0.001 and **** *p* ≤ 0.0001. Bars represent means of triplicate measurements and error bars represent standard deviation.

**Figure 2 toxins-14-00229-f002:**
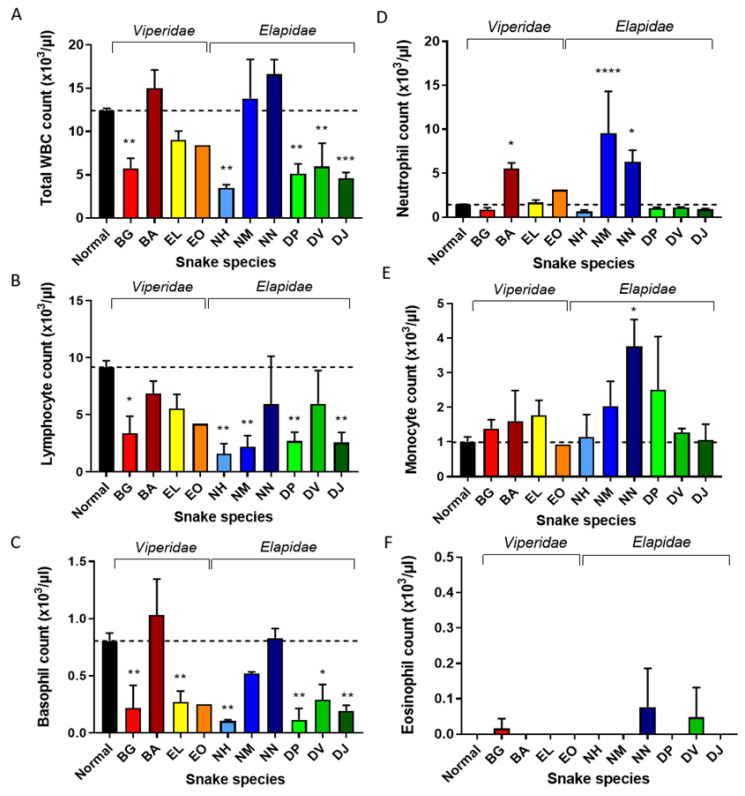
Snake species-specific effects of venom on total WBC counts and specific cell types. Thin blood films from mice subjected to venoms from ten African snakes were examined at ×40 magnification to complete (**A**) total white blood cell (WBC) counts, and at ×100 magnification to complete absolute counts of (**B**) neutrophils, (**C**) lymphocytes, (**D**) monocytes, (**E**) basophils and (**F**) eosinophils (number of cells per μL blood). Venoms: BG (*B. gabonica*), BA (*B. arietans*), EL (*E. leucogaster*), EO (*E. ocellatus*), NH (*N. haje*), NM (*N. melanoleuca*), NN (*N. nigricollis*), DP (*D. polylepis*), DV (*D. viridis*), DJ (*D. jamesoni*). Parameters from envenomed groups were compared to those from the control group (Normal: injected with PBS) with Dunnett’s multiple comparison test. Significant values are indicated by * *p* ≤ 0.05, ** *p* ≤ 0.01, *** *p* ≤ 0.001 and **** *p* ≤ 0.0001. Bars represent means of duplicate counts and error bars represent standard deviations.

**Figure 3 toxins-14-00229-f003:**
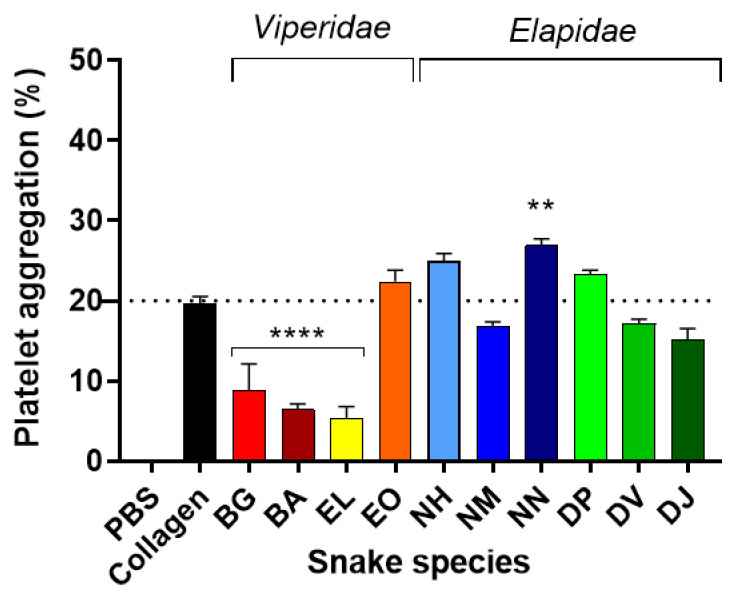
The effect of African snake venoms on human platelet aggregation. Using PBS as a negative control (0% aggregation) and collagen as a positive control (mean 20%), platelet aggregation of ex vivo human platelets was quantified after treatment with the following venoms: BG (*B. gabonica*), BA (*B. arietans*), EL (*E. leucogaster*), EO (*E. ocellatus*), NH (*N. haje*), NM (*N.*
*melanoleuca)*, NN (*N. nigricollis*), DP (*D. polylepis*), DV (*D. viridis*), DJ (*D. jamesoni*). The resulting data were statistically analysed with Dunnett’s multiple comparison test. Significant values are indicated by ** *p* ≤ 0.01 and **** *p* ≤ 0.0001. Bars represent means of duplicate counts and error bars represent standard deviations.

**Figure 4 toxins-14-00229-f004:**
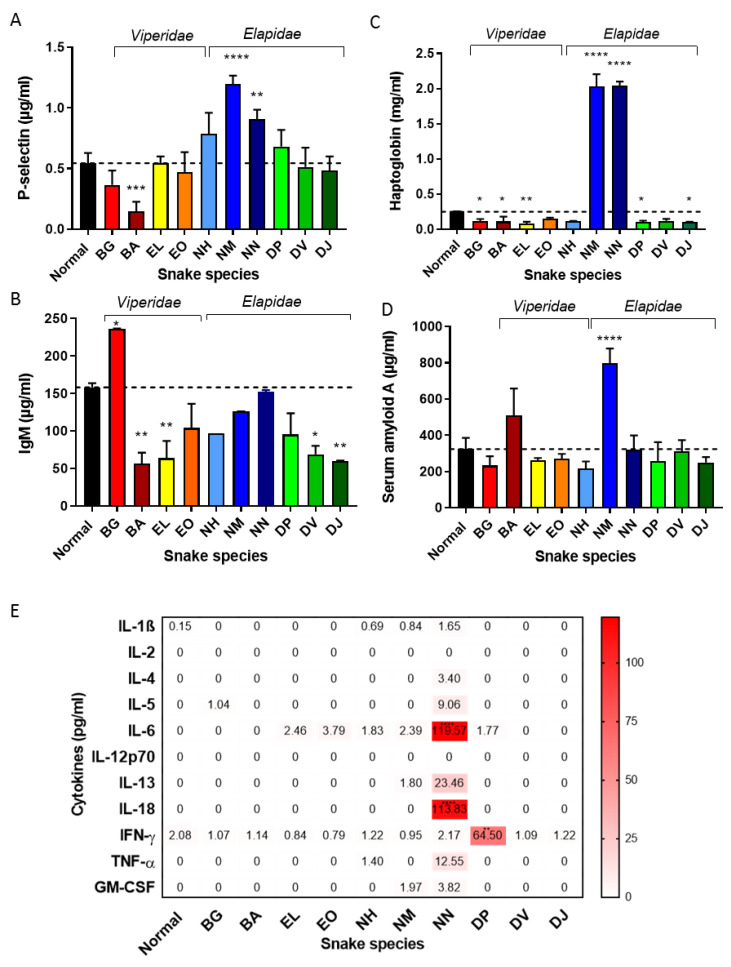
African snake venoms elicit variant acute phase and inflammatory responses in the systemic circulation of mice. Sera from mice six hours post-injection of ten snake venoms were assessed for levels of (**A**) P-selectin; (**B**) IgM; (**C**) haptoglobin, (**D**) SAA and (**E**) cytokines (darker red indicates high elevation, pink indicates moderate elevation and white indicates minor or no change; see colour legend) after treatment with the following venoms: BG (*B. gabonica*), BA (*B. arietans*), EL (*E. leucogaster*), EO (*E. ocellatus*), NH (*N. haje*), NM (*N. melanoleuca*), NN (*N. nigricollis*), DP (*D. polylepis*), DV (*D. viridis*), DJ (*D. jamesoni*). Values found to be significant compared to the normal control (PBS only) are indicated by * *p* ≤ 0.05, ** *p* ≤ 0.01, *** *p* ≤ 0.001 and **** *p* ≤ 0.0001. Data represent means of duplicate measurements and error bars in (**A**–**D**) represent standard deviations.

**Figure 5 toxins-14-00229-f005:**
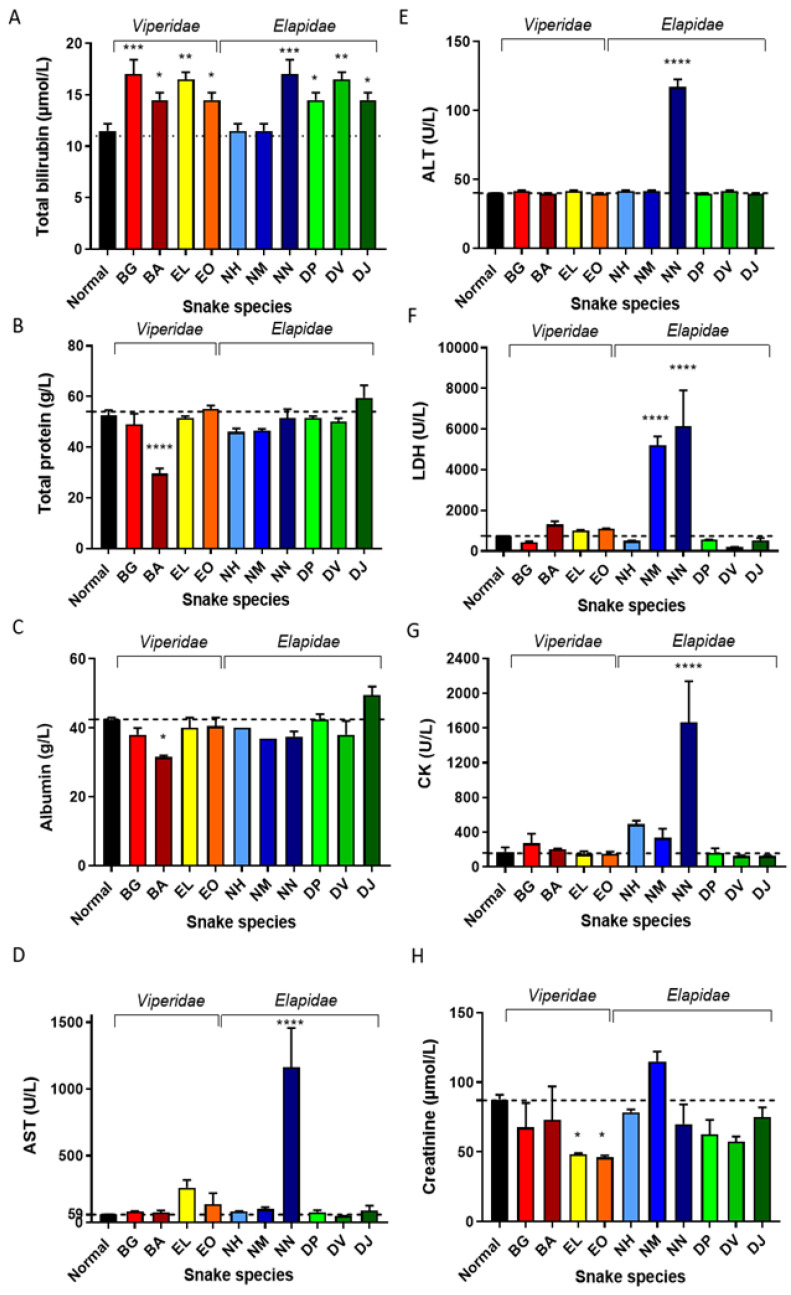
Serum biochemistry parameters suggest acute hepatic and renal damage in mice injected with *N. nigricollis* venom. An automated clinical chemistry analyser was used to measure (**A**) total bilirubin; (**B**) total protein; (**C**) albumin; (**D**) AST; (**E**) ALT; (**F**) LDH; (**G**) CK and (**H**) creatinine in sera of mice injected with the following venoms: BG (*B. gabonica*), BA (*B. arietans*), EL (*E. leucogaster*), EO (*E. ocellatus*), NH (*N. haje*), NS (*N. melanoleuca*), NN (*N. nigricollis*), DP (*D. polylepis*), DV (*D. viridis*), DJ (*D. jamesoni*). Values found to be significant compared to the normal control (PBS inoculation only) are indicated by * *p* ≤ 0.05, ** *p* ≤ 0.01, *** *p* ≤ 0.001 and **** *p* ≤ 0.0001. Bars represent means of duplicate measurements and error bars represent standard deviations.

**Table 1 toxins-14-00229-t001:** Grading of platelet aggregation in mice injected with medically important snake venoms. Platelet aggregation was recorded according to relative size and distribution of platelet colonies per field of view (100× objective): (i) no aggregation, <1 small colony/field; (ii) mild aggregation, 2–5 colonies/field and/or small colonies only; (iii) moderate aggregation, 2–8 colonies/field and/or small and medium colonies; (iv) marked aggregation, >5 colonies/field and/or medium-large colonies present. Small, medium and large colonies were defined as consisting of 2–10 platelets, 11–20 platelets and >20 platelets, respectively.

Snake Species	Platelet Aggregation				Comments
	No	Mild	Moderate	Marked	
**Normal**		+			Mild platelet aggregation (2 colonies/field, small colonies)
** *B. gabonica* **	-				No platelet aggregation
** *B. arietans* **	-				No platelet aggregation
** *E. leucogaster* **		+			Mild platelet aggregation (2 colonies/field, small-medium colonies)
** *E. ocellatus* **		+			Mild/No platelet aggregation (<1 colonies/field, small colonies)
** *N. haje* **		+			One small (2–8 platelets/colony) colony per field
** *N. melanoleuca* **				+++	Marked platelet aggregation (3–4 colonies/field, small, and medium colonies)
** *N. nigricollis* **			++		Moderate platelet aggregation (>2 colonies/field, small and medium colonies)
** *D. polylepis* **			++		Mild platelet aggregation (2–3 small colonies/field, medium and large colonies)
** *D. viridis* **		++			Mild platelet aggregation (3–4 colonies/field, small colonies)
** *D. jamesoni* **				+++	Marked platelet aggregation (3–4 colonies/field, small, and medium colonies)

**Table 2 toxins-14-00229-t002:** Patterns of acute responses to the venom of African viper (*B. gabonica* (BG); *B. arietans* (BA); *E. leucogaster* (EL); *E. ocellatus* (EO)) and elapid (*N. haje* (NH); *N. melanoleuca* (NM); *N. nigricollis* (NN); *D. polylepis* (DP); *D. viridis* (DV); *D. jamesoni* (DJ)) snakes identified from results of human and murine assays performed in this study. The departures of these responses from normal values (green) are illustrated as a heat map and (+, ++, +++, and ++++) represent the relative extent of increases or decreases.


		Normal		Low/Mild		High/Moderate		High ^+++^/Marked		Low ^++/+++/++++^	
Marker of Venom Induced:	Assay(s)	Venoms from *Echis* and *Bitis* Viper Species				Venoms from *Naja* and *Dendroaspis* Elapid Species(#—Spitting Cobra)					
		BG	BA	EL	EO	NH	NM	NN^#^	DP	DV	DJ
**Human assays (ex vivo)**											
**Changes to human blood parameters**	Haemoglob-in	**Low**	**Low ^++++^**	**Normal**	**Low**	**Low ^++++^**	**Low**	**Low ^++++^**	**Low ^++++^**	**Low ^++++^**	**Low ^++++^**
	Haemolysis	**Normal**	**Normal**	**Normal**	**Normal**	**Normal**	**Normal**	**Marked**	**Normal**	**Normal**	**Normal**
	Platelet aggregation	**Low**	**Low ^++++^**	**Low ^++++^**	**High ^+^**	**High ^++^**	**Low**	**High ^+++^**	**High ^++^**	**Low**	**Low**
**Murine assays (in vivo)**											
**Changes to blood cell counts, morpholo-gy, and platelets**	Platelet aggregation	**None**	**None**	**Mild**	**Mild**	**Mild**	**Marked**	**Moderate**	**Moderate**	**Mild**	**Marked**
	RBC morphology	**Normal**	**++**	**Normal**	**Normal**	**Normal**	**Normal**	**+**	**Normal**	**Normal**	**+**
	Total WBC	**Low ^++^**	**High**	**Low**	**Low**	**Low ^++^**	**High**	**High**	**Low ^++^**	**Low ^++^**	**Low ^++^**
	Lymphocyt-es	**Low ^+^**	**Low**	**Low**	**Low**	**Low ^++^**	**Low ^++^**	**Low**	**Low ^++^**	**Low**	**Low ^++^**
	Basophils	**Low ^++^**	**High**	**Low ^++^**	**Low**	**Low ^++^**	**Low**	**Normal**	**Low ^++^**	**Low ^+^**	**Low ^++^**
	Neutrophils	**Normal**	**High ^+^**	**Normal**	**Normal**	**Normal**	**High ^++++^**	**High ^+^**	**Normal**	**Normal**	**Normal**
**Activation of acute phase and inflammat-ory responses**	P-selectin	**Low**	**Low ^+++^**	**Normal**	**Normal**	**High**	**High ^++++^**	**High ^++^**	**High**	**Normal**	**Normal**
	IgM	**High ^+^**	**Low ^++^**	**Low ^++^**	**Low**	**Low**	**Low**	**Normal**	**Low**	**Low ^+^**	**Low ^++^**
	Haptoglob-in	**Normal**	**Normal**	**Normal**	**Normal**	**Normal**	**High ^++++^**	**High ^++++^**	**Normal**	**Normal**	**Normal**
	Serum Amyloid A	**Low**	**High**	**Low**	**Low**	**Low**	**High ^++++^**	**Normal**	**Low**	**Normal**	**Low**
	Cytokines	**Normal**	**Normal**	**Normal**	**Normal**	**Normal**	**Normal**	**IL6, ^++^ IL18** **TNF-α, IL13**	**IFNg ^++^**	**Normal**	**Normal**
**Damage to kidney**	Serum Creatinine	**Low**	**Low**	**Low ^+^**	**Low ^+^**	**Low**	**High**	**Low**	**Low**	**Low**	**Low**
**Damage to liver function**	Total serum protein	**Normal**	**Low ^++++^**	**Low**	**Normal**	**Low**	**Low**	**Normal**	**Normal**	**Normal**	**High**
	Serum Albumin	**Normal**	**Low ^+^**	**Normal**	**Normal**	**Normal**	**Low**	**Low**	**Normal**	**Normal**	**High**
	Serum Bilirubin	**High ^+++^**	**High ^+^**	**High ^++^**	**High ^+^**	**Normal**	**Normal**	**High ^+++^**	**High ^+^**	**High ^++^**	**High ^+^**
	Serum ALT	**Normal**	**Normal**	**Normal**	**Normal**	**Normal**	**Normal**	**High ^++++^**	**Normal**	**Normal**	**Normal**
**Damage to muscle**	Serum CK	**Normal**	**Normal**	**Normal**	**Normal**	**Normal**	**Normal**	**High ^++++^**	**Normal**	**Normal**	**Normal**
	Serum LDH	**Normal**	**Normal**	**Normal**	**Normal**	**Normal**	**High ^++++^**	**High ^++++^**	**Normal**	**Normal**	**Normal**
	Serum AST	**Normal**	**Normal**	**High**	**Normal**	**Normal**	**Normal**	**High ^++++^**	**Normal**	**Normal**	**Normal**

## Data Availability

The data presented in this study are available in this article and Appendix A.

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
