# Peer review of "Profiling the Murine Acute Phase and Inflammatory Responses to African Snake Venom: An Approach to Inform Acute Snakebite Pathology"

_toxins, 2022, doi:10.3390/toxins14040229_

Round 1
Reviewer 1 Report
The work of these colleagues is very interesting because it once again shows the diversity of pathophysiological phenomena involved in snakebites. I was very sensitive to the desire to apply the 3Rs. My only remark would be to add within the limits the small number per group which may be the cause of a lack of sensitivity on certain biological anomalies.
Author Response
We thank the reviewer for their careful reading of the manuscript and their constructive remarks. Our response follows (the reviewer comments are in italics).
The work of these colleagues is very interesting because it once again shows the diversity of pathophysiological phenomena involved in snakebites.
Author response:
Thank for very much for your comments.
I was very sensitive to the desire to apply the 3Rs. My only remark would be to add within the limits the small number per group which may be the cause of a lack of sensitivity on certain biological anomalies.
Author response:
We would like to thank the reviewer for the comments. We are very limited with the numbers of animals in the experiment and per group due to ethical and financial constraints, combined with this representing a pilot study. We agree that this is a limitation and can certainly impact upon our interpretations where parameters may vary extensively between animals. In response to the reviewer’s point, we have further emphasised this in the limitation statement made in the conclusion of the manuscript.
Reviewer 2 Report
The authors measured inflammatory responses to African snake venoms in murine model since they might be relevant for new diagnostic and treatment approaches. Therefore, it fits the journal scope in my opinion. The introduction seems too long, like it is written in the review form, especially the part describing detailly so far gained results of other research groups. My major concern is that the phenomenon observed in mice don’t necessarily have to reflect those occurring in human victims, since not every species is sensible to snake venom toxins in the same way. Array of performed ex vivo assays on human blood is too poor (only platelet aggregation, haemoglobin content and haemolysis) and I don’t see the purpose of their performance. The authors should discuss the fact that for more straightforward conclusions the further research should be pushed towards clinical observations.
In 5.2. there is a sentence “At the end of the experiment, the number of surviving mice injected with one venom LD50 dose (the amount of venom resulting in the survival of 50% of the injected mice; see Supplementary Table S2) were selected for sample collection to provide samples of comparable severity of systemic envenoming...” According to my knowledge, in lethal toxicity assay, the groups receive a range of different venom doses, and the result is the one that causes death in 50% of the animals. There is rarely a clear-cut situation when a group gets a certain venom amount that kills half of the tested subjects and you know that the remaining ones from the same group got one LD50 dose. That’s why Spearman Karber method is usually used for calculation. It is unclear how the authors identified survived animals that received exactly one LD50 dose for sample collection 6 h following challenge in order to achieve comparability? Maybe I’m missing something here and would be very grateful if the authors could provide an explanation. Also, 6 h seems as too short period for some pathological alterations to occur. Why there were no additional sampling points?
In 2.1.1. result concerning D. jamesoni is associated with Figure 1C, not 2C.
In 2.1.2. Figure 1F is mentioned before Figure 1E.
In the caption of Figure 1 under “A” normal red blood cell morphology is presented, not abnormal as stated (“Abnormal murine red blood morphology (A-D)”).
In 2.3.3. and 2.2.4 titles every species should be written in italic (as well as ex vivo throughout the manuscript).
Author Response
We would like to thank the reviewer for the careful and thorough reading of this manuscript and for the thoughtful comments and constructive suggestions, which help to improve the quality of this manuscript. Our detailed responses to each point follow below.
The authors measured inflammatory responses to African snake venoms in murine model since they might be relevant for new diagnostic and treatment approaches. Therefore, it fits the journal scope in my opinion.
Author response:
Thank for very much for your supportive comments.
The introduction seems too long, like it is written in the review form, especially the part describing detailly so far gained results of other research groups.
Author response:
We thank the reviewer for providing their thoughts here, although we prefer to retain the introduction in its current form. It represents two pages, which certainly is not unreasonable for a publication, and while we appreciate the point about the ‘review context’ we feel this is important because of the scarcity of studies on this topic published in the literature to date. We hope that the reviewer understands our position on this.
My major concern is that the phenomenon observed in mice don’t necessarily have to reflect those occurring in human victims, since not every species is sensible to snake venom toxins in the same way.
The reviewer of course makes a viable point here, but until now there is no alternative assays to replace preclinical assays. Clearly, such preclinical research can inform future clinical research strategies to investigate/validate such findings.
Array of performed ex vivo assays on human blood is too poor (only platelet aggregation, haemoglobin content and haemolysis) and I don’t see the purpose of their performance. The authors should discuss the fact that for more straightforward conclusions the further research should be pushed towards clinical observations.
Author response:
We would like to thank the reviewer for the comments. In this study we compare some pathological findings from mouse models to ex vivo human blood models in the hope of reducing the use of experimental animals in future research. While we agree that the conclusions from these research activities are mixed, we believe that retaining them still holds value for the manuscript and that certain parameters enable integration between ex vivo human experiments and in vivo animal experiments – e.g. Section 2.1.1 where RBC abnormalities suggest venom-induced haemolysis in envenomed mice, and ex vivo findings, suggesting certain venoms caused damage to red blood cells, resulting in anaemia and spherocytosis and the loss of haemoglobin to the extracellular space. However, we do of course agree that the greatest value of this research moving forwards would be via clinical research, and we have added a sentence to the conclusion section of the manuscript to emphasise this point.
Reviewer comment 5: In 5.2. there is a sentence “At the end of the experiment, the number of surviving mice injected with one venom LD50 dose (the amount of venom resulting in the survival of 50% of the injected mice; see Supplementary Table S2) were selected for sample collection to provide samples of comparable severity of systemic envenoming...” According to my knowledge, in lethal toxicity assay, the groups receive a range of different venom doses, and the result is the one that causes death in 50% of the animals. There is rarely a clear-cut situation when a group gets a certain venom amount that kills half of the tested subjects and you know that the remaining ones from the same group got one LD50 dose. That’s why Spearman Karber method is usually used for calculation. It is unclear how the authors identified survived animals that received exactly one LD50 dose for sample collection 6 h following challenge in order to achieve comparability? Maybe I’m missing something here and would be very grateful if the authors could provide an explanation. Also, 6 h seems as too short period for some pathological alterations to occur. Why there were no additional sampling points?
Author response:
We thank the reviewer for this point and apologise that it is not clear. They are completely correct that the animal samples analysed did not equate to exactly one LD50 dose. Experimental animals were dosed with different amounts of venom and then the LD50 calculated, and samples collected from the group of experimental animals that received the dose of venom CLOSEST to the calculated LD50 were used. The methods section of the manuscript has been updated to correct this error in the previous submission, and the venom doses that the mice received and that were analysed in this paper are displayed in Supplementary Table 2. We agree that the 6 h timepoint may mean that we miss some pathological alterations, and this is highlighted in the conclusions of the manuscript as a limitation that needs addressing in future work, however, due to the experimental design (i.e. using experimental animals being dosed for 6 hr LD50 purposes), we were unable to generate additional time point samples for use in this particular study.
In 2.1.1. result concerning D. jamesoni is associated with Figure 1C, not 2C.
Author response:
Thanks for spotting this - we have corrected this error in the revised manuscript.
In 2.1.2. Figure 1F is mentioned before Figure 1E.
Author response:
This has been corrected in the revised manuscript.
In the caption of Figure 1 under “A” normal red blood cell morphology is presented, not abnormal as stated (“Abnormal murine red blood morphology (A-D)”).
Author response:
This has been corrected in the revised manuscript.
In 2.3.3. and 2.2.4 titles every species should be written in italic (as well as ex vivo throughout the manuscript).
Author response:
This has been corrected in the revised manuscript.
Round 2
Reviewer 2 Report
The authors have argumentatively replied to my comments and explained their choices which I accept, together with the changes in the new version of the manuscript, introduced according to the provided suggestions. All clear now.